# Analysis of the Contact Area for Three Types of Upper Limb Strikes

**DOI:** 10.3390/jfmk7020050

**Published:** 2022-06-17

**Authors:** Vaclav Beranek, Petr Stastny, Frederic Turquier, Vit Novacek, Petr Votapek

**Affiliations:** 1Department of Rehabilitation Fields, Faculty of Health Care Studies, University of West Bohemia, 301 00 Pilsen, Czech Republic; vnovacek@ntc.zcu.cz; 2Department of Sport Games, Faculty of Physical Education and Sport, Charles University, 162 52 Prague, Czech Republic; stastny@ftvs.cuni.cz; 3Ecole Spéciale de Mécanique et d’Electricité, Sudria, 69002 Lyon, France; frederic.turquier@esme.fr; 4Biomechanical Human Body Models, New Technologies—Research Centre, University of West Bohemia, 301 00 Pilsen, Czech Republic; 5Department of Machine Design, Faculty of Mechanical Engineering, University of West Bohemia, 301 00 Pilsen, Czech Republic; pvotapek@kks.zcu.cz

**Keywords:** combat sports, self-defense, Mixed Martial Art, direct punch, elbow strike, palm strike, hand, injury

## Abstract

Performance in strike combat sports is mostly evaluated through the values of the net force, acceleration, or speed to improve efficient training procedures and/or to assess the injury. There are limited data on the upper limb striking area, which can be a useful variable for contact pressure assessment. Therefore, the aim of this study was to determine the contact area of the upper limb in three different strike technique positions. A total of 38 men and 38 women (*n* = 76, 27.3 ± 8.5 years of age, 73.9 ± 13.8 kg of body weight, 173.3 ± 8.4 cm of body height) performed a static simulation of punch with a fist, palm strike, and elbow strike, where three segments of the right upper limb were scanned. The analysis of 684 images showed a correlation (r = 0.634) between weight and punch technique position in men and significant differences in elbow strike (*p* < 0.001) and palm strike (*p* < 0.0001) between women and men. In both groups, the palm demonstrated the largest area and the elbow the smallest one. These data may be used to evaluate strike contact pressure in future studies in forensic biomechanics and assessment of injury in combat sports and self-defense.

## 1. Introduction

Studies analyzing net force [1,2,3,4,5,6,7,8,9,10,11], acceleration [12,13,14,15], and velocity [16,17,18,19] for strikes by limbs are very important for the assessment of striking-oriented martial arts such as Boxing, Kickboxing, Muay Tai, Karate, and Taekwondo [20], and self-defense for police, military, and civilian personnel [21,22,23]. Explosive strength and dynamics of movements are prerequisites for high performance in a wide range of striking disciplines [24,25], where the punch is a key component [26]. Its use is in self-defense without any protective equipment and according to the type of strike also in specific combat sports such Kyokushin karate [27], Mixed Martial Art [28], Muay Thai [29]. After the impact of the punch, the force is applied for a short period of time [30] (less than 50 ms with time to peak 5.08 ± 0.57 ms) [31] as the product of high mass and acceleration just before the impact [32]. The speed before the impact is the decisive factor determining the impact force [33,34]. It is useful to know other variables, such as the striking area of the upper limb, for pressure calculation, potentially leading to alternative head injury assessment compared to the standard peak force criterion [35]. Previous studies [30,32] show that, in addition to the standard variables, the description of the strike area is not sufficiently addressed even though it may have a major impact on the interpretation of interactive forces and pressures developed in strikes.

In the field of combat sports, monitoring of the foot pressure data or center of pressure (COP) has been reported in a few studies [16,36], and, compared to that, there are not enough reports on the pressure caused by the impact of the upper limb on the target. In the field of ergonomics, fall, and forensic biomechanics, the upper limb in connection with pressure was described to better estimate the risks of injury for different areas of the human body [37]. The COP displacement on the surface of the fist during the punch and striker grip strength was reported in connection with a hand injury in boxers [38] and competitors of martial arts [39]. Furthermore, hand morphology relates to performance in various areas of sport [40,41] and determines the handgrip strength [42,43,44]. Therefore, the report on the hand contact area has a high potential to inform on the impact characteristics of the strike [32], where these effects have not yet been investigated.

Besides strength and size, the skinfold thickness of the hand affects the severity of the impact [45] where a smaller contact area increases the maximum stress on the target [32]. Compared to that, the larger contact area results in the lower contact pressure and material strain, thereby higher fracture thresholds have to be assumed in the case of elastic contact characteristics due to the larger contact area [37]. Thus, the thickness of the muscles and fat on the palm is an important factor depending on the sex [46], where women have a higher proportion of body fat [47,48] than men and lower muscle mass [49]. Physical exercise causes asymmetrical development of bone density mass, size, and musculature of the dominant limb [50], and the asymmetry in palm size correlates with handedness [51]. These factors can play important role in the compressive response that affects the maximum impact force [46]. The striking surface of a fist is less than 60% of the area of the whole palm. This means that if the total force applied in a strike is the same, the stress in the targeted tissue will be 1.7–3.0 times greater in a punch by a fist than in a palm strike [32]. However, more precise information about the effect of the striking hand area on the pressure value is missing, and if so, it comes from other fields. Pressure on the ulnar area and force are greater and may cause injury more easily during the resuscitation [52]. Compared to that, the highest force occurs on the trapezium and scaphoid bone during a fall onto outstretched hands in snowboarding, skiing, bicycle racing, in-line skating, ice skating, and certain gymnastics/acrobatics maneuvers [53]. The reported results show the need to compare both groups of men and women, as well as individual types of strikes according to different strike areas.

Since there are limited data about the size of the upper limb striking area, the main goal of this study is to determine the 2D nondeformable area (mm^2^) of the three parts of the upper limb used in the straight punch by fist, the straight strike by palm, and elbow strike in men and women.

## 2. Materials and Methods

### 2.1. Study Design

This cross-sectional study was performed during one familiarization session and one testing session separated by 48 h, where both sessions had the same schedule from 8 AM to 15 PM. Participants were measured for their body part contact areas during the simulation of the three types of strikes (straight punch with fist, straight strike by open palm, straight strike by elbow). The subjects received a detailed explanation from a highly skilled operator on how to achieve a correct static position of the upper limb during the measurement. The participants performed 3 static simulations of the straight punches with a clenched fist, 3 straight strikes with an open palm, and 3 strikes by the elbow (olecranon) in a randomized order and as close as possible to the position of the upper limb during the real impact on the target. Visual inspection was performed by the operator.

### 2.2. Participants

A total of 38 men and 38 women (*n* = 76, 27.3 ± 8.5 years of age, 73.9 ± 13.8 kg of body weight, 173.3 ± 8.4 cm of body height) practicing self-defense at an advanced level for 6 to 12 months and older than 18 years with no injuries or other medical restrictions signed informed consent about the purpose and content of the study. Body height was measured by Velleman WM10050 (Velleman, Belgium); body weight was measured by a Personal scale ETA Vital Body 6780 (HP Tronic, Czech Republic). The study protocol was approved by the local ethical committee of the Faculty of Physical Education and Sport, Charles University, Prague, Czech Republic (No. 267/2019), and was in accordance with the Declaration of Helsinki (2013). The heights, weights, and ages of subjects enrolled in the study are summarized in Table 1.

### 2.3. Procedures of Data Collection

In a standing position, each subject performed a static simulation of a punch with a fist (Figure 1a), palm strike (Figure 1b), and elbow strike (Figure 1c), where the impact phase simulation on the sensor board lasted 5 s and three segments of the right upper limb were photographed separately at a height of 105 cm. Each subject exerted a slight pressure on the plate during the image recording under the supervision of a highly skilled operator. The subject performed 3 imprints for each position, i.e., 3 clenched fist imprints, 3 open palm imprints, and 3 elbow imprints, interrupting the contact with the transparent plate between each trial. The contact area for the clenched fist was the metacarpal joint area and the proximal phalanges from the dorsal side with the thumb pulled to the distal and middle phalanges without contact with the transparent plate; the angle of the forearm to the plate was approximately 90 degrees. The contact area for the palm strike was the carpal and metacarpal area, where the fingers were not in contact with the transparent plate, and the angle of the forearm to the plate was approximately 50 degrees. The contact area for the elbow strike was the olecranon, the arm in the flexion position; the angle of the forearm to the plate was approximately 30 degrees.

### 2.4. Instrumentation and Data Acquisition

All simulations of strikes were performed on 2D optical contact area scanner Podocam (ING Corporation, s.r.o., Frýdlant nad Ostravicí, Czech Republic, Figure 1, [42,43,44]), reported in the previous study [54] where a static record of a selected segment of the human body was made. The device consists of an all-metal aluminum frame with fixation to the board, total size of 38 cm × 37 cm; and a camera holder, 1 Full HD web camera Logitech HD Pro Webcam C920, Romanel-sur-Morges, Switzerland; special transparent plates; mirror; LED light sources 230 V, 50 Hz, 8 W; and original software. A special foil was installed in selected parts of the frame to avoid unwanted light effects. During the experiment, there was no other light source except the technical red-light source with a power of 10 watts. Reduced lighting conditions required longer camera exposure. The fixed height of the scanning plate enabled stability during photography and a fixed fixation of the device.

### 2.5. Image Postprocessing

MATLAB^®^ R19b (The MathWorks^®^, Inc., Natick, MA, USA) version was used to postprocess the acquired images. The process is illustrated step by step in Figure 2 for all three techniques:(a)The image was loaded and cropped to the relevant region of interest (same for all images).(b)The image was converted into CIELAB color space. The first channel was extracted and normalized to 16-bit grayscale and represented as a matrix.(c)The components of the matrix were squared to increase the contrast and normalized again to a 16-bit grayscale.(d)A logical mask was created using a 40% contrast threshold.(e)Clusters smaller than 50 pixels were removed.(f)Possible holes in the mask were filled.

**Figure 2 jfmk-07-00050-f002:**
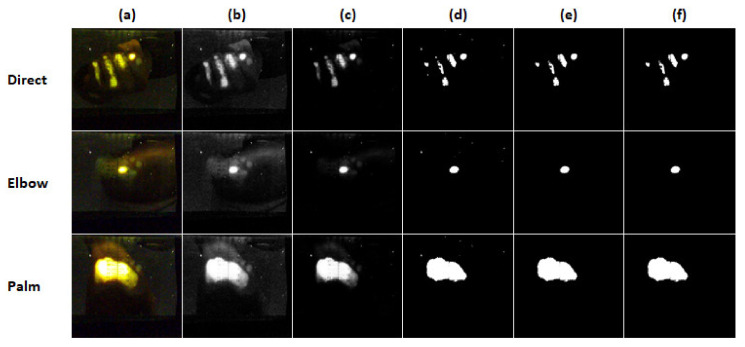
Illustration of image postprocessing for all three techniques. (**a**) The image is loaded and cropped to the relevant region of interest (same for all images). (**b**) The image is converted into CIELAB color space. The first channel is extracted and normalized to 16-bit grayscale and represented as a matrix. (**c**) The components of the matrix are squared to increase the contrast and normalized again to 16-bit grayscale. (**d**) A logical mask is created using a 40% contrast threshold. (**e**) Clusters smaller than 50 pixels are removed. (**f**) Possible holes in the mask are filled.

The resulting mask is a matrix of logical values with dimensions corresponding to the resolution of the cropped image. Each matrix component corresponds to one square pixel. The sum of all logical values equal to 1 represents the area of the masked object in square pixels. The area was converted to square millimeters using the Podocam scale visible on the image where 10 mm equals 34 pixels in our setting. The area was converted to equivalent diameter d considering the area to be a disk, i.e., d = √4 A/π, where A is the area of the mask in square millimeters.

### 2.6. Statistical Analysis

The independent variables were the sex and types of the impact area of strikes (straight punch with fist, straight strike by open palm, straight strike by elbow). The dependent variable was the equivalent diameter d (mm). With power 0.8 and α = 0.05 and based on preliminary data, the sample size was estimated to 37 for strike comparison and 19 for subgroups analyses.

Variability in the three repetitions for each subject and strike technique was assessed using the percent relative range criterion C defined as the range of the three measured values divided by their average and multiplied by 100. This criterion allows for identifying subjects that did not perform the same technique consistently. Intraclass correlation coefficient ICC assessed the degree to which individual performances resemble each other [55]. A MATLAB^®^ R19b function ICC from 2008 by Arash Salarian [56] was used for this purpose. Correlation between the equivalent diameter and age, weight, and height was assessed in Excel 2016. All further statistical tests were conducted in MATLAB^®^ R19b. Lilliefors test assessed data normality. ANOVA assessed the effects of sex and technique. Student’s *t*-test was used as a post hoc test to compare data between women and men and between techniques in case of normality; Wilcoxon signed-rank test was used otherwise. All tests were performed at 5% significance level where relevant.

## 3. Results

The results in terms of equivalent diameter are shown in Figure 3 and summarized in Table 2. Expressed in terms of equivalent diameter replacing the imprint by a disk of the same area, direct strike demonstrated imprint diameter from 11.7 mm to 55.3 mm, elbow strike from 6.0 mm to 33.4 mm, and palm strike from 27.6 mm to 67.2 mm.

The only correlation between equivalent diameter and weight was found in men and direct technique with r = 0.634. No other correlation between equivalent diameter and height, weight, or age was found.

Relative range criterion C was higher than 20% in only 19 out of 228 image triplets suggesting good repeatability of the measurement within the population of volunteers. High variability was observed 7 times in the direct technique, 11 times in the elbow technique, and once in the palm technique. High variability results are illustrated in Figure 4 showing how differently a subject can perform in three consecutive trials. ICC was as good as 0.9852 with the confidential interval from 0.9816 to 0.9882 indicating that the values from the same triplet of images tend to be similar.

Equivalent diameter had normal distribution for women in all three techniques. In men, the equivalent diameter was not normally distributed. ANOVA showed a significant effect of the technique and interaction term with *p* < 0.001. Wilcoxon signed-rank test comparing women and men showed a significant difference in equivalent diameter in Elbow and Palm strike technique with *p* < 0.001 and *p* < 0.0001, respectively, as shown in Figure 5. In women and men groups separately, the difference between techniques (all combinations of pairs of techniques) was significant, as evident from Figure 5.

## 4. Discussion

The main goal of study was to determine the area of the three parts of the upper limb: the straight punch by fist, the straight strike by palm, and the elbow strike. The palm strike technique imprint demonstrated the largest area, and the elbow strike technique imprinted the smallest one. In terms of equivalent diameter, there was no significant difference between women and men in punch strike technique imprints. Upon contact of the clenched fist with the target in the perpendicular direction, the contact area is represented by the exposed distal interphalangeal joints, proximal interphalangeal joints, metacarpophalangeal joints, and carpometacarpal joints from the dorsal side, where there is no other significant muscle or fat tissue. Compared to that, the contact area of the elbow (primary olecranon) is also exposed to part of the ulnar side of the forearm, but there is a significant portion of the muscle and fat tissue that may interfere during the strike. The palm is formed, apart from the skeletal base, by four muscle groups, some of which form the surface relief of the palm (especially the thenar muscles on the radial side and the hypothenar muscles on the ulnar side). Because body fat and musculature are highly plastic with the greatest changes under the influence of training and/or diet compared to skeletal tissue [57], the contact area of the fist will not be as affected by environmental factors, as in the palm and elbow. 

In palm strike techniques, the difference in equivalent diameter was significant between women and men with a higher average value in the men’s group. This may be attributed to generally larger palms in men, who have significantly longer metacarpals and phalanges than women [58] and longer average hand length and breadth of a palm [59,60]. In contrast, a difference of elbow strike technique imprints in equivalent diameter was also significant between women and men but with a higher average value in the women’s group. This may be related to more subcutaneous tissue in women within the assessed population, or in general [47,48,49].

The only correlation between equivalent diameter and weight was found in men and direct technique. This is not confirmed by previous studies, where no correlation was found between the body weight and length of index fingers. Conversely, a significant correlation has been found between the length of the index finger and height [61] or hand length and height [62].

The triplets acquired for each subject and technique showed in most cases low variations in terms of area and equivalent diameter. High variability results occurred in only 19 out of 228 recorded image triplets suggesting good repeatability of the measurement within the population of volunteers. It is hypothesized that there are several ways to execute each strike technique, and it may affect the repeatability. In a straight punch with a fist, the imprint area depends on the position of the distal part of the metacarpal bones and the proximal digit from the dorsal side during contact with the target, where the preference of the radial or ulnar side is decisive. This conclusion is the same for the palm strike technique, where the influence of radial and ulnar preferences for forces and pressures under the palm has been previously reported [52,53]. In the elbow strike technique, the imprint area can vary from a small size imprint of the olecranon to a large area of the forearm, due to the degree of flexion and pronation of the forearms. Increasing flexion exposes the ulnar protrusion (olecranon) in the target, reducing contact of surrounding tissue with the target and thereby decreasing the contact area. The study results can already enable assessing the risk of fracture based on reported bone tolerance values [63,64], for example, of the temporal bone when submitted to a static punch simulation involving the fist, palm, or elbow as in our experiments. Such conditions could occur in mixed martial art combat or in self-defense when an individual without any training in self-defense or martial arts and without external help would want to keep immobilized an aggressive individual laying down on the ground by applying some constraints similar to pressure point techniques on the lateral side of the head, as it offers the largest surface. In such a case, the temporal bone appears to be of great interest as it is the thinnest bone of the skull [65] and therefore at the highest risk. It exhibits a rather upper anterior flat zone that can be approximated by a plane. However, its double-layer structure made of a bone plate covered by soft tissues (mainly fat and skin on the outer surface and meninges on the inner surface) differs significantly from the glass plate used in our experiments. As far as contact areas are concerned, experimental results may underestimate equivalent reality while, as a consequence, for a given force, experimental pressure assessment may be overestimated. Provided that subject A can only engage his upper body in the loading process, the force applied on the temporal bone could be estimated around two-thirds of his body weight [66]. In our study, this would translate into peak forces of 440 N and 540 N on average, respectively, for women and men. As a result, estimated contact pressure for men (women) would reach, on average, 0.3 (0.3) MPa, 0.6 (0.5) MPa, and 2.5 (1.6) MPa for the palm, direct, and elbow loading modalities, respectively. According to the probability of fracture vs. peak pressure (worked out form the peak force) relationship already established [67], the average man elbow static punch simulation could lead to the fracture of the upper temporal bone in around 1 case out of 20. The risk of fracture for the remaining modalities could be considered as negligible. Even without dealing with impact but just dynamic conditions such as for the gait, the peak pressure could increase four-fold [68] and lead to moderate and high risk of fracture for both fist and elbow punch conditions, respectively. All these numbers are preliminary estimates that need to be confirmed by further investigations as peak pressures and percent of body weight engaged may be overestimated and body weights are average numbers. However, the recommendations could already be made not to use the elbow but favor the palm in an immobilization maneuver involving the head.

### Limitations

The main limit of this study can be upper limb position during scanning. The blows/strikes have a high variability of execution both in the system of self-defense and, for example, in MMA, because it is possible to execute them from the stance as well as during ground fighting. The position of the hand during the scan is a compromise where the contact area and the angle of the forearm to the impact plane is crucial. In addition, selected height of scan and vertical position during scanning was chosen with regard to scan stability and subject comfort. A next limitation of the study can be the selection of the right upper limb to determine the striking area. We expected it to be the preferred limb by the athletes of mixed martial arts (MMA), who use an elbow without the presence of protective elements, where the vast majority (80.3%) of MMA fighters reported using an orthodox stance with the right dominant hand [69]. Besides that, the difference between right- and left-hand lengths were not significant in both sexes [59].

In the next limit, it is necessary to consider the difference between experimental conditions to the real situation. The “impact” area of the measurement device (Podocam) was a planar compared to the real situation where the strike and impact on human tissue will be exposed. Likewise, the impact area was rigid, while in the real situation, different directions of target movement can be expected. Moreover, the pressure distribution is not yet known and is likely to be inhomogeneous.

## 5. Conclusions

The values of the upper limb striking area expressed as equivalent diameter of a disc pointed out differences between the individual strike techniques. Straight punches by a fist, palm, and elbow are an effective tool for dominance over the opponent, but they should also be considered in injury risk assessment. The methodology of force measurement for three types of strikes is reported [30,63], and a future study will focus on the evaluation of strike pressure in the area of forensic biomechanics, self-defense, and injury risks assessment in combat sports that do not use protective equipment for selected strike areas.

## Figures and Tables

**Figure 1 jfmk-07-00050-f001:**
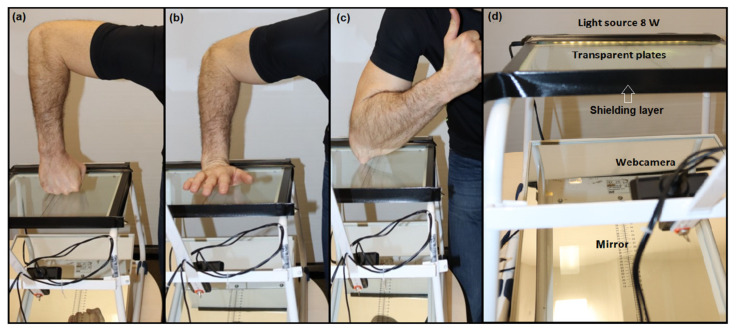
The body position for (**a**) straight punch, (**b**) palm strike, (**c**) elbow strike contact area on the measuring platform and (**d**) overall view.

**Figure 3 jfmk-07-00050-f003:**
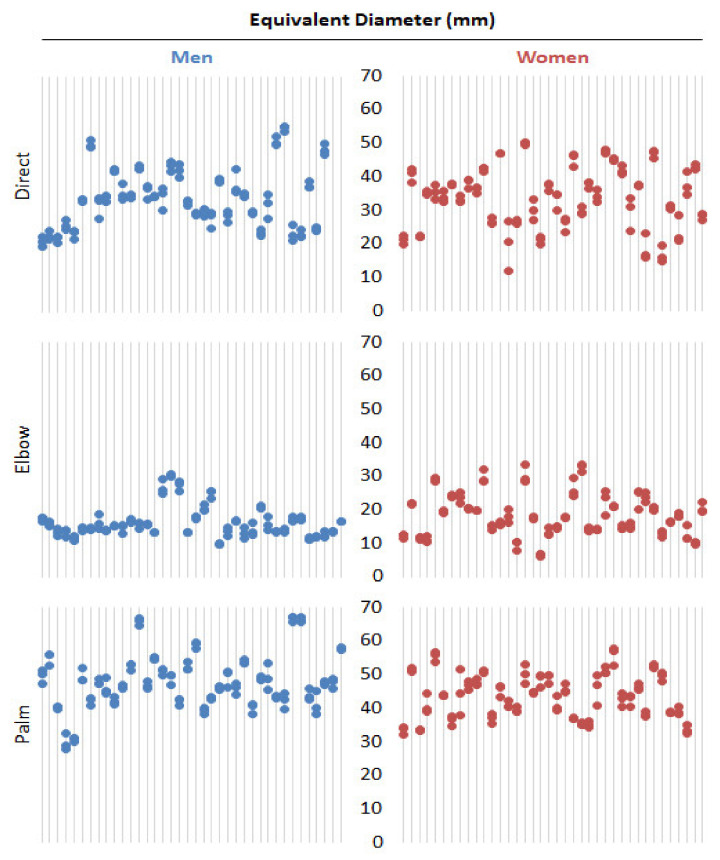
Equivalent diameter (mm) dot plots for women and men groups and all three techniques.

**Figure 4 jfmk-07-00050-f004:**
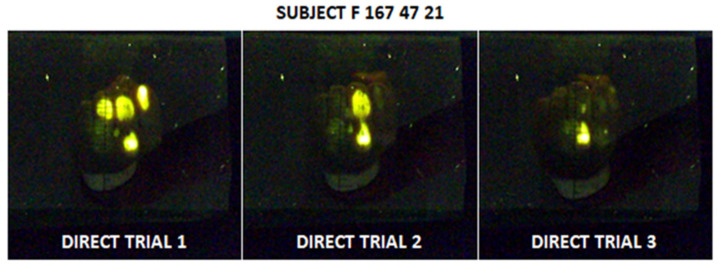
Illustration of variability in three consecutive trials for selected subject and direct technique.

**Figure 5 jfmk-07-00050-f005:**
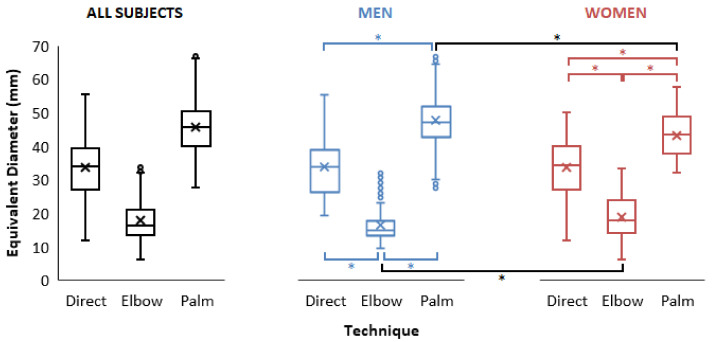
Equivalent diameter (mm) boxplot with median indicated by × symbol for all subjects, women and men, for all three strike techniques. Statistically significant differences are indicated with an asterisk symbol *.

**Table 1 jfmk-07-00050-t001:** Summary of heights, weights, and ages of subjects. Average ± standard deviation (SD) and range (minimum-maximum) are shown.

	Men (*n* = 38)	Women (*n* = 38)	All Subjects (*n* = 76)
Age (years)	27.9 ± 7.0 (19–43)	26.6 ± 9.8 (19–52)	27.3 ± 8.5 (19–52)
Weight (kg)	81.5 ± 13.8 (60–110)	66.2 ± 8.8 (47–89)	73.9 ± 13.8 (47–110)
Height (cm)	178.2 ± 7.7 (160–198)	168.4 ± 5.8 (158–178)	173.3 ± 8.4 (158–198)

**Table 2 jfmk-07-00050-t002:** Summary of Equivalent Diameter (mm) for different techniques. SD= standard deviation.

Technique	Subjects	Mean ± SD (Minimum-Maximum)
Direct	All	33.8 ± 8.9 (11.7–55.3)
	Men	33.7 ± 8.8 (19.5–55.3)
	Women	33.8 ± 9.0 (11.7–50.2)
Elbow	All	17.2 ± 5.6 (6.0–33.4)
	Men	16.0 ± 4.6 (9.5–30.6)
	Women	18.4 ± 6.3 (6.0–33.4)
Palm	All	45.8 ± 7.7 (27.6–67.2)
	Men	47.9 ± 8.2 (27.6–67.2)
	Women	43.7 ± 6.5 (32.2–57.7)
All	All (*n* = 76)	33.8 ± 8.9 (11.7–55.3)

## Data Availability

The data is available on request from corresponding author.

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
