# Peer review of "Analysis of the Contact Area for Three Types of Upper Limb Strikes"

_jfmk, 2022, doi:10.3390/jfmk7020050_

Round 1

Reviewer 1 Report

In the study, the contact areas in three different types of strikes are compared. 

The methodology is sound and the results well presented. I suggest to the authors the following amendments:

1) when trying to interpret the results with respect to a real-world situations, three main limitations must be considered 

- the target was planar (as opposed to most body parts)

- the target was rigid (as opposed to body parts)

- the force distribution within the contact area is not known and probably far from homogenous (for example, in palm strikes most force is going to be applied through the palm near the wrist and the fingers are going to contribute only a little) 

Some of the above limitations were mentioned in the discussion, I would recommend to expand the discussion and/or limitations section.

2) The description of the contact area in fist punches (Lines 209 - 211) is anatomically incorrect (there are no proximal, intermediate and distal finger joints and metacarpal joints; there are distal interphalangeal joints, proximal interphalangeal joints, metacarpophalangeal joints and carpometacarpal joints); please correct

3) The description of variations as "satisfactory" (Line 230) is not suitable; I suggest "in most cases low" or similar.

4) The example described at the end of the discussion section ist not realistic - to push a fist or an ellbow against the side of the head of a lying person is hardly the way how any healthcare professional (trained in self-defense or not) would try to immobilize a patient. If at all, such actions could occurr in a brawl.

After the above suggested revisions I consider the paper suitable for publication.

Author Response

Response to Reviewer 1 Comments

In the study, the contact areas in three different types of strikes are compared.

The methodology is sound and the results well presented. I suggest to the authors the following amendments:

Point 1: when trying to interpret the results with respect to a real-world situations, three main limitations must be considered 

 - the target was planar (as opposed to most body parts)

 - the target was rigid (as opposed to body parts)

 - the force distribution within the contact area is not known and probably far from homogenous (for example, in palm strikes most force is going to be applied through the palm near the wrist and the fingers are going to contribute only a little) 

Some of the above limitations were mentioned in the discussion, I would recommend to expand the discussion and/or limitations section.

Response 1:

Thank you for the recommendation to add the limits of the study. Now, from line 295, other study limits are listed:

“In the next limit, it is necessary to consider the difference of experimental conditions compared to the real situation. The "impact" area of the podocam was a planar compared to the real situation where the strike and impact human tissue will be exposed. Likewise, the impact area of the podocam was rigid, while in the real situation, different directions of target movement can be expected. Moreover, the pressure distribution is not yet known and is likely to be inhomogeneous.”

Point 2: The description of the contact area in fist punches (Lines 209 - 211) is anatomically incorrect (there are no proximal, intermediate and distal finger joints and metacarpal joints; there are distal interphalangeal joints, proximal interphalangeal joints, metacarpophalangeal joints and carpometacarpal joints); please correct

Response 2: Thank you for specifying the anatomical name, which we have modified according to the recommendations in line 221.

Point 3: The description of variations as "satisfactory" (Line 230) is not suitable; I suggest "in most cases low" or similar.

Response 3: Thank you for the recommendation. On line 242, we used a more appropriate term.

Point 4: The example described at the end of the discussion section ist not realistic - to push a fist or an ellbow against the side of the head of a lying person is hardly the way how any healthcare professional (trained in self-defense or not) would try to immobilize a patient. If at all, such actions could occurr in a brawl.

Response 4: Thank you for an important note regarding the described self-defense movement. We believe that pressure points are part of the technical tools of self-defense and as such it is possible to use these specific movement maneuvers to overcome the resistance of the enemy. "Pressure actions" are used mainly in combat on the ground, where a tactical coach can include it in training. We have changed the description to determine the specific defensive response in line 260.

Reviewer 2 Report

Dear Authors:

First of all, congratulations for the work done. You have developed a correct research but with errors that should be solved.

ABSTRACT:
The r representing the correlation factor should be in lower case.

INTRODUCTION:
I miss the reference to aquatic sports in which the prevalence of upper limb injuries is high (e.g. DOI: 10.3390/jcm10050902).
The previous hypothesis could be added at the end of this section.

METHODS:
Ensure that all abbreviations are defined the first time they appear in the text and in all tables (in Table 1 SD and Y are not described, for example).
The statistical analysis should be complemented, at least, with the calculation of the effect sizes of the statistical tests applied.

Kind regards

Reviewer 3 Report

Overview

The authors aimed to determine the contact area of the upper limb in three different strike technique positions in combat sports.

The topic is interesting in the field of strike combat sports.

I have some suggestions for authors to improve the manuscript.

Specific comments

Abstract

-Lines 19-20: Please add at least one decimal number to the mean and standard deviation.

Introduction

The authors did a good job of synthesizing the literature despite having cited many previous studies.

The introduction follows the logic from what is known to what is not known; the gap to be filled is clear and the purpose is relevant.

Materials and Methods

The methodology is clear, and the measurements were carried out objectively. Measurement instruments are validated, and measurements have been tested for reliability.

The statistics used are appropriate.

-Question: Was an a priori power analysis carried out to determine the sample size? It is always recommended to do an analysis to first determine the number of subjects to be recruited for the study.

To fix:

-Line 80: Replace with a simpler 'Study Design'.

-Line 91: Please add at least one decimal number to the mean and standard deviation.

-Table 1:

Replace with 'Women (n=38) and Men (n=38)'.

If you indicate men first and then women in the text (abstract and Participants), you should do so throughout the manuscript. Either change it in the text or in the table.

Please put age, weight, and height in order (from top to bottom).

Replace with 'Age (years)'.

Please add at least one decimal number to the mean and standard deviation.

-Figures:

Figure captions must be placed below the figure.

Results

-Line 192: Standard Deviation is denoted by SD for convention. If necessary, insert notes below the table to add information.

-Indicate in the table for "All, Men and Women" the number of subjects. The tables should be independent of the text and readers should not have to re-read the text to interpret the tables.

- Figure captions must be placed below figures 3 and 4.

Discussion

-Rewrite the first paragraph by beginning with the purpose of the study and then describe concisely and clearly what was found and what is new.

Conclusions

The take-home message is clear.

The authors' conclusions are justified.

Round 2

Reviewer 3 Report

Thank you for your replies.